# Two Year Study of Aflibercept and Ranibizumab Intravitreal Therapy in Patients with Wet AMD

**DOI:** 10.3390/medicina57121293

**Published:** 2021-11-24

**Authors:** Dorota Luksa, Anna Heinke, Katarzyna Michalska-Małecka

**Affiliations:** 1University Clinical Center, University Hospital Medical University of Silesia, 40-514 Katowice, Poland; dorotaluksa@gmail.com (D.L.); annaheinke@gmail.com (A.H.); 2Department of Ophthalmology, School of Medicine in Katowice, Medical University of Silesia, 40-751 Katowice, Poland

**Keywords:** age related-macular degeneration, wet AMD, aflibercept, ranibizumab, intravitreal injections, choroidal neovascularization, OCT

## Abstract

*Background and objectives:* The aim of this study was to evaluate the therapeutic results in patients with exudative AMD treated with ranibizumab and aflibercept intravitreal injections over a two-year observation period. *Materials and methods:* A retrospective observational study was conducted in a clinical hospital on a group of patients who randomly qualified for treatment with Aflibercept (group A) and Ranibizumab (group B) as part of the Polish National Health Fund Medical Program for exudative AMD. Group A consisted of 90 patients, and group B contained 54 patients. The choice of drug in a patient depended solely on the availability of the medication at the time. Before each injection, best corrected visual acuity (BCVA) on the ETDRS scale and central retinal thickness (CRT) were assessed using optical coherence tomography (OCT). Patients from both groups were treated in the first year of treatment with a rigid scheme of 3 doses of 2.0 mg Aflibercept (group A) and 0.5 mg Ranibizumab (group B) at monthly intervals, followed by 4 doses at bimonthly intervals. In the second year, a “pro re nata” scheme was applied. The aim was to evaluate changes in BCVA and CRT after three injections, after 7 injections (about 12 months), and after the second year of therapy (24 months) with reference to the baseline and to compare the effectiveness of the medications. The influences of the following factors were studied: age, gender, initial BCVA, and initial CRT, as well as the number of injections received. *Results*: No significant statistical differences were found between patients receiving Aflibercept and Ranibizumab therapy in terms of achieving improved visual acuity and reducing retinal thickness after two years of therapy. *Conclusions*: Both aflibercept and ranibizumab were found to be effective for treating exudative AMD.

## 1. Introduction

Age-related macular degeneration (AMD) affects the elderly population and leads to a loss of central vision. It is manifested by degenerative changes in the central part of the retina.

The most important factor influencing the development of AMD is age. Other established causes are smoking, diabetes, alcohol abuse, and hereditary factors. Factors with a possible influence on the development of the disease include having a high level of C-reactive protein or a low level of antioxidants, dyslipidemia, and light colour of the iris. Relationships with exposure to chemical agents, such as lead and iron, and with long-term exposure to radiation have also been shown (wavelength 400–550 nm, close to UV and blue light) [1].

Ranibizumab (Lucentis^®^) and Aflibercept (Eylea^®^) are antivascular endothelial growth factor drugs that have been approved for the treatment of age-related macular degeneration (AMD) [2]. Ranibizumab was approved in the United States in 2006 and entered the European market in 2007 after approval by the European Medicines Agency (EMA). Aflibercept was introduced in 2011 and 2012 in the USA and Europe, respectively [3]. Results of the key studies ‘ANCHOR’ and ‘MARINA’ for Ranibizumab [4,5] and ‘VIEW 1’ and ‘VIEW 2’ for Aflibercept [3,6] confirmed the effectiveness of both drugs for improving and stabilizing visual acuity and preventing vision loss in patients with exudative AMD.

Almost every patient requires long-term therapy and numerous injections in order to achieve this therapeutic effect. The following treatment schemes have been developed: rigid, “as needed” (“Pro Re Nata”, PRN), and “Treat and Extend”. The rigid scheme is based on injections administered at the same intervals that are not subject to modification. “Pro Re Nata” requires follow-up visits to determine whether the patient has the characteristics of disease activity and, therefore, requires additional injections. Relapse of the exudative form of the disease can be deduced from the presence of hemorrhages in the central retina by ophthalmoscopic examination as well as from the presence of sub- and/or intraretinal fluid in the central retina, assessed via OCT examination. It can also be confirmed by optical coherence tomography angiography of the macula (neovascular membrane with measurement of its size, with evaluation of vascular loop morphology and density, quantitative flow assessment, flow density map) or fluorescein/indocyanine green angiography (contrast leakage).

The interval between injections in the “Treat and Extend” scheme is subject to appropriate lengthening or shortening depending on AMD activity, and the aim is to determine the interval at which no relapse takes place.

In our study, we compared the results of 2-year Aflibercept and Ranibizumab therapy in clinical practice in patients who received rigid-scheme injections in the first year of treatment, followed by a pro re nata scheme in the second year.

## 2. Materials and Methods

This retrospective observation study was conducted in a clinical hospital. Observations were conducted on a group of patients who qualified randomly for treatment with Aflibercept (group A) and Ranibizumab (group B) in the Medical Program for the treatment of wet AMD within the National Health Fund from January 2017 to March 2018. According to the criteria set on 1 January 2017, the following requirements had to be met in order to be accepted to the Program: (1) the presence of active (primary or secondary), classical, occult, or mixed subretinal neovascularization (CNV) occupying more than 50% of AMD confirmed with optical coherent tomography (OCT) and fluorescein angiography or optical coherent tomography angiography (angio-OCT); (2) age ≥ 45; (3) lesion size of less than 12 optic nerve disc areas (12 DA); (4) best corrected visual acuity (BCVA) in the treated eye of 0.2–0.8, as determined according to the Snellen charts (or ETDRS equivalent 50–80 letters, as appropriate); (5) patient’s consent for intravitreal injections; (6) absence of dominant geographical atrophy; and (7) absence of dominant hemorrhage. The exclusion criteria were (1) hypersensitivity to Aflibercept, Ranibizumab, or any of the excipients; (2) active infection of the eye; (3) pregnant or breastfeeding; (4) occurrence of side effects associated with the drug, preventing its further use; (5) rhematogenous retinal detachment or grade 3 or 4 macular hole; (6) lack of cooperation from the patient; and (7) progression of the disease defined as deterioration of the best corrected visual acuity (BCVA) to <0.2 determined according to the Snellen charts (less than 20 letters in ETDRS equivalent) lasting longer than 2 months or the presence of permanent damage to the foveal structure, making it impossible for the patient to obtain stability or functional improvement.

In the second year of treatment (‘pro re nata’ regimen), disease activity was detected at follow-up visits based on ophthalmoscopy and the results of optical coherence tomography of the macula.

Group A (Aflibercept) consisted of 90 patients, including 50 women and 40 men aged between 57 and 89 years. The average age was 69.51 years. Group B (Ranibizumab) consisted of 54 patients, including 26 women and 24 men aged 52 to 96 years. The average age was 77.48 years.

In the case where a given patient had both eyes in the treatment program, data from therapy of the right eye were included in the analysis. During the two-year follow-up period, 17 patients receiving Aflibercept were excluded from the study: 6 people who regularly missed the scheduled follow-up visits for unjustified reasons, 9 people with progression of the changes and a decrease in visual acuity below V < 0.2 lasting for two subsequent follow-up visits, and 2 deaths. The treatment outcomes of these individuals were not considered in the statistical analysis of the study. The study group of patients receiving Aflibercept after exclusion included 90 subjects. All patients receiving Ranibizumab remained in the study. Patients were administered 2 mg/0.05 mL of Aflibercept and 0.5 mg/0.05 mL of ranibizumab intravitreally.

Before each injection, the best corrected visual acuity (BCVA) was assessed using Snellen charts, and scores were then converted to the ETDRS scale for statistical purposes. Appropriate conversion tables were used for this purpose. Central retinal thickness (CRT) was also evaluated using optical coherence tomography (XR OCT Avanti RTVue XR system). All patients from both groups received 7 intravitreal injections in the first year of treatment: 3 doses each of 2.0 mg Aflibercept (group A) and 0.5 mg Ranibizumab (group B) at monthly intervals, followed by 4 doses at bimonthly intervals. In the second year of treatment, therapy with the abovementioned drugs was carried out using the “pro re nata” treatment regimen (additional intravitreal injections given in the case of disease activity features at follow-up visits, which took place about every 6–7 weeks). For group A, the average number of injections given during the pro re nata period of treatment was 3.03, while for group B, it was 3.59. The analysis with the Mann–Whitney U-range test showed that between-group differences in the number of injections in the second year of treatment were not statistically significant (U-2036; *p* > 0.05).

The aim was to evaluate changes in best corrected visual acuity and central retinal thickness on the first day of therapy, after three injections (third month), after 7 injections (about 12 months), and after the second year of therapy (24 months) and compare the effectiveness of the medications. The influences of the following factors on the conducted observations were studied: age, gender of the patient, initial best corrected visual acuity, initial central retinal thickness, and the number of injections received.

## 3. Statistical Analysis

The compliance of the distributions of the main variables with the normal distribution was checked. Due to the lack of compliance, non-parametric tests were used. The Wilcoxon test was used to determine relationships within groups. To compare the parameters of the groups to each other, the Mann–Whitney U test was used.

## 4. Results

One hundred and forty-four eyes from 144 patients who received anti-VEGF injections into the vitreous body chamber were analyzed over a period of two years as part of the AMD National Health Fund Treatment Programme. This included 90 patients receiving Aflibercept (group A) and 54 patients receiving Ranibizumab (group B). Characteristics of the groups in terms of age, gender, initial visual acuity, and initial central retinal thickness are presented in Table 1, Table 2, Table 3 and Table 4. The analysis with the Mann–Whitney rank test showed that differences between groups A (Aflibercept) and B (Ranibizuman) in age, gender, initial visual acuity, and initial retinal thickness were not statistically significant.

The main aim was to evaluate changes in visual acuity and central retinal thickness on the first day of therapy, after three injections (third month), after 7 injections (about 12 months), and after 2 years of therapy (24 months) and compare the effectiveness of both medications. In group A, after the first year of treatment, an improvement in visual acuity was observed in 73.33% of patients (66 patients), stabilization was achieved in 21.11% of patients (19 patients), and deterioration occurred in 5.56% of patients (5 patients).

In group A, the initial visual acuity was 59.9 letters on the ETDRS scale, while after one year of treatment, it was 69.2 letters. The average improvement was 9.3 letters (statistically significant in the Wilcoxon test, *p* < 0.01). In group B, after the first year of therapy, an improvement in visual acuity was observed in 44.3% of patients (24 patients), stabilization was achieved in 42.6% of patients (23 patients), and deterioration to less than 15 ETDRS letters occurred in 13.1% of patients (7 patients). The initial visual acuity was 60.7 letters, while after one year of treatment, it was 65.0 letters. The average improvement was 4.3 letters (statistically significant in the Wilcoxon test: −3.4; *p* < 0.01). The analysis showed that the differences between group A and group B in terms of changes in visual acuity after the first year of treatment were statistically significant (Mann–Whitney U test; U = 1688; *p* < 0.01), and the difference was higher in group A.

Another analyzed endpoint was the change in central retinal thickness after the first year of treatment. Initially, in group A, the mean central retinal thickness was 370.4 μm, and after one year, it decreased to 300.7 μm. In group B, these values were 358.2 μm and 300.7 μm, respectively. These results were statistically significant (*p* < 0.01) in the individual groups (Wilcoxon test). On the other hand, overall decreases in central retinal thickness compared with baseline values did not differ significantly statistically between groups (Mann–Whitney U test; U = 2004; *p* > 0.05). The greatest decrease in retinal thickness in both groups occurred at the fourth control point (i.e., after the third injection), followed by a gradual, small increase in retinal thickness in the central area. The increases in central retinal thickness during the second year of treatment compared to the first year of treatment were not statistically significant. For group A, the average decrease was about 7 µm (from 300.7 to 307.9 µm), and for group B, it was about 11 µm (from 311.7 to 322.7 µm).

In group A, after the second year of therapy, visual acuity improved in 47.0% of patients (42 persons) compared with baseline values before therapy started. Stabilization occurred in 24.0% of patients (22 persons), and deterioration occurred in 27% of patients (24 persons). After the second year of treatment, the average visual acuity was 65.4 letters. The average improvement was 5.5 letters (statistically significant in the Wilcoxon test: *Z* = −7.96; *p* < 0,01). Changes in visual acuity for patients being treated with Aflibercept in the two-year follow-up are shown in Figure 1. In group B, after the second year of therapy, visual acuity improved in 40.6% of patients (22 patients) compared with baseline values before therapy. Stabilization occurred in 44.4% of patients (24 patients), and deterioration occurred in 15% of patients (8 patients). After the second year of treatment, the average visual acuity was 64.4 letters. 

The average improvement in visual acuity was 3.7 letters (statistically significant in the Wilcoxon test: *Z* = −2.96; *p* < 0.01). Changes in visual acuity for patients treated with Ranibizumab over many years of follow-up are presented in Figure 2. The analysis showed that the differences between group A and group B in terms of changes in visual acuity after the second year of treatment were not statistically significant. It is worth noting that the result in terms of visual acuity after one year of Aflibercept therapy was objectively higher. Unfortunately, this effect significantly reduced in the second year of treatment. After the first year, Ranibizumab therapy was associated with an improvement in visual acuity, but the effect was lower compared with that produced with Aflibercept therapy. However, the effect achieved in this group after one year of treatment was also maintained in the second year of treatment. The comparison of Aflibercept and Ranibizumab treatments is presented in Figure 3.

Almost no groups showed a correlation between gender or age and the best corrected visual acuity after one and two years of therapy. For group A, the mean ETDRS after 7 injections was found to be higher among women (M = 71.3; SD = 8.35) than among men (M = 67.27; SD = 12.7). The Mann–Whitney rank-sum test analysis showed that the differences between groups were statistically significant (U = 697.5; *p* < 0.05). ETDRS scores after 2 years showed no significant differences between women and men (U = 678; *p* > 0.05). Moreover, no correlation was found between the patient’s age and the visual acuity achieved after one year of therapy (Spearman test, correlation coefficient of 0.5). There was a moderate negative correlation (rho = −0.38; *p* < 0.001) between patient age and ETDRS after two years of treatment; that is, the higher the age, the lower the ETDRS values. For group B, no statistically significant differences were found between men and women in terms of ETDRS after 7 injections and in terms of ETDRS after 2 years in the Mann–Whitney test (U = 337; *p* > 0.05 and U = 358.5; *p* > 0.05, respectively). There was also no correlation between patient age and best corrected visual acuity after 1 or 2 years of treatment. Spearman’s test produced correlation coefficients of −0.10 and −0.12, respectively.

The changes in retinal thickness after the second year of treatment were analyzed (Figure 4a for Aflibercept and Ranibizumab). In group A, the mean thickness of the central retina was 307.96 μm, while in group B, it was 322.70 μm, and these values were significantly lower than the baseline values (*p* < 0.01; Wilcoxon test). It is worth noting, however, that, in group A, the central retinal thickness variation curve was more stable over time, while, in group B, this parameter increased after the first and second years of therapy. However, the final results did not differ significantly between groups (Figure 4a,b).

In addition, appropriate subgroups were created in each studied group (A and B) depending on the initial ETDRS visual acuity (first subgroup: 35–50 letters; second subgroup: 51–70 letters; third subgroup: 71–85 letters). For patients receiving Aflibercept, the first subgroup included 27 patients, the second subgroup included 47 patients, and the third subgroup included 16 patients. In patients treated with Ranibizumab, there were, respectively, 26, 14, and 14 patients. For both therapies, the highest increase in ETDRS letters took place in the first subgroup, i.e., in patients with the lowest initial visual acuity; however, for patients treated with, Aflibercept, this occurred at the 8th control visit (after one year of therapy), whereas for Ranibizumab, visual acuity improvement was lower and was noted at the 4th control visit (after third injection). The therapeutic effect in this subgroup was more stable for Ranibizumab after 2 years of therapy. The visual acuity curve in patients belonging to the intermediate subgroup was similar to that of the first subgroup. In the subgroup with the best initial visual acuity, this parameter decreased after the second year of therapy for both drugs (these dependencies are shown in the Figure 5 for Aflibercept and Figure 6 for Ranibizumab).

## 5. Discussion

Anti-VEGF intravitreal therapy stabilizes and often improves visual acuity in patients with exudative AMD. No significant statistical differences were found between patients receiving Aflibercept and Ranibizumab therapy in terms of achieving improved visual acuity and reducing retinal thickness after two years of therapy. Similar observations have been obtained in numerous clinical trials. [3,7] The specialist literature contains information on the equality of these therapies after one year of treatment [3,8,9,10,11], which differs from our observations. Differences in observations may be due to the size of the study group and also the frequency of drug administration. The analysis showed that the differences between groups A and B in terms of changes in visual acuity after one year of therapy compared with baseline values were statistically significant, and the difference was significantly higher in the Aflibercept group. It should be noted that in 9 patients receiving Aflibercept, it was necessary to terminate treatment under the drug program for exudative AMD due to significant progression of lesions. In the group receiving Ranibizumab, this situation did not occur, but the study group was smaller. It is worth noting that both these groups received exactly the same number of injections (seven) in the first year of treatment. The increase in ETDRS letters in the Aflibercept-treated group was 9.29, higher than that usually given in the scientific literature (5.3 letters increase with an average number of injections of 7.1) [12], although similar growth, e.g., 8 letters with an average number of injections of 8.7, can be seen in some sources [13]. Among patients treated with Ranibizumab, the average letter gain in ETDRS after one year of therapy was 4.35 letters. Similar values can be found in a study that evaluated the results of anti-VEGF therapy in clinical practice (increase of 4.24 letters with 5.88 injections in the first year of treatment) [12]. Kumar and co-authors studying the effect of Aflibercept therapy in patients who were previously treated unsuccessfully with Ranibizumab and also presented more satisfactory results after the use of Aflibercept [14]. Similar conclusions were also reached by Heussen and co-authors. After analyzing their results, they concluded that Aflibercept therapy may be an effective method of therapy in patients with resistance to Ranibizumab treatment [15]. An extensive meta-analysis of therapy results of 26.360 patients obtained from 42 observational studies conducted in clinical practice (published between 2007 and 2015) showed average increases of 8.8 letters (system treat and extend, average 7.3 injections/year) and 3.5 letters (pro re nata system, average 5.4 injections/year) in the first year of Ranibizumab therapy [16].

The greatest decrease in central retinal thickness in both treatments occurred after the third intravitreal injection. The same result was presented in a study conducted in Switzerland on 102 patients (112 eyes) receiving Aflibercept [13]. In addition, the patients experienced a slight increase in retinal thickness after an initial decrease, but this was noticeable from about 21 months of therapy (an increase of about 15 μm at the end of a two-year therapy compared to the lowest values obtained). In the case of Aflibercept, in our study, an increase occurred from the 12th month of therapy, but the increase was small, with an average increase of 8 μm achieved after the end of the two-year therapy period. In patients receiving Ranibuzmab, a steady but small increase was already visible after the fourth month, leading to an eventual increase of about 20 μm. For both anti-VEGF agents, there was a statistically significant decrease in the thickness of the central retina in comparison with baseline values.

It is worth considering whether a higher frequency of injections would allow us to achieve better therapy results. This especially applies to the second year of pro re nata therapy—Aflibercept in our study. In other scientific publications, similar conclusions showed that the use of more frequent injections improves visual acuity. Researchers have reported that a 5-letter improvement in visual acuity for Aflibercept requires 5–6 injections on average, while for Ranibizumab, 6–7 injections are required, on average [12]. It should be taken into account that this would involve a significant reduction in the intervals between control visits to almost double the number of visits in the second year of therapy. The desired effect would be to maintain the balance between the timing of therapy, the inconvenience of treatment for the patient (high frequency of control visits), and finally, the cost of treatment and the real benefits of therapy. There are reports about the superiority of the “treat and extend” treatment scheme over the “pro re nata” system [16,17]. It allows a balance between the appropriate timing between injections. It optimizes treatment, as each patient is approached individually. This system reduces the number of follow-up visits required and, thus, reduces the overall cost of treatment.

## 6. Conclusions

Both Aflibercept and Ranbizumab are effective for treating exudative age-related macular degeneration, which translated into an improvement in visual acuity and a reduction in central retinal thickness over a two-year observation period. After the first year of therapy, visual acuity improved in both groups, but it was significantly higher in the Aflibercept group. The therapeutic effect, obtained after one year of therapy was more well maintained after the second year of therapy in the group receiving Ranibizumab (therapy stability). The greatest potential for improvement in visual acuity occurs in patients with good initial visual acuity. A higher frequency of intraocular injections is associated with better therapy results being achieved by patients. In the first year of therapy, both groups of patients (7 injections/year) achieved significant benefits. The second year of therapy was conducted using the “pro re nata” system, and the frequency of injections decreased by about half. An optimal therapy system should be considered, so that patients can achieve the best possible results and maintain the achieved therapeutic effect for as long as possible.

## Figures and Tables

**Figure 1 medicina-57-01293-f001:**
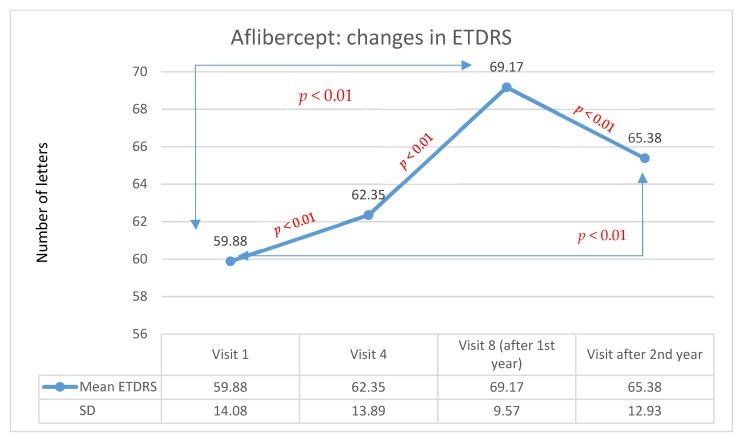
Changes in visual acuity in the ETDRS during individual follow-up visits for the Aflibercept group. Red indicates statistically significant differences (Wilcoxon test).

**Figure 2 medicina-57-01293-f002:**
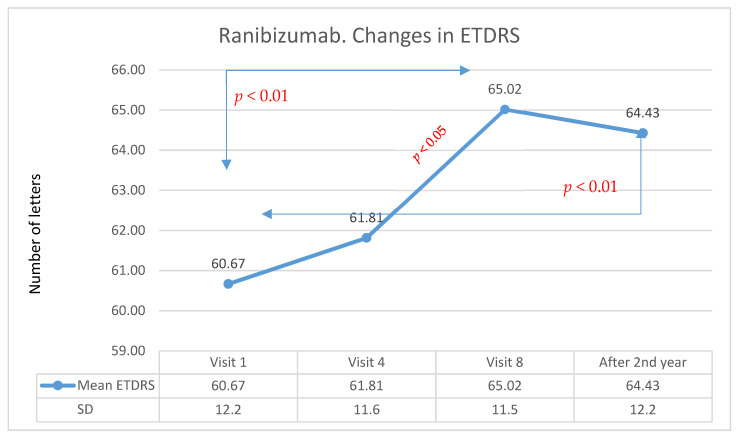
Changes in visual acuity in the ETDRS during individual follow-up visits for the Ranibizumab group. Red indicates statistically significant differences (Wilcoxon test).

**Figure 3 medicina-57-01293-f003:**
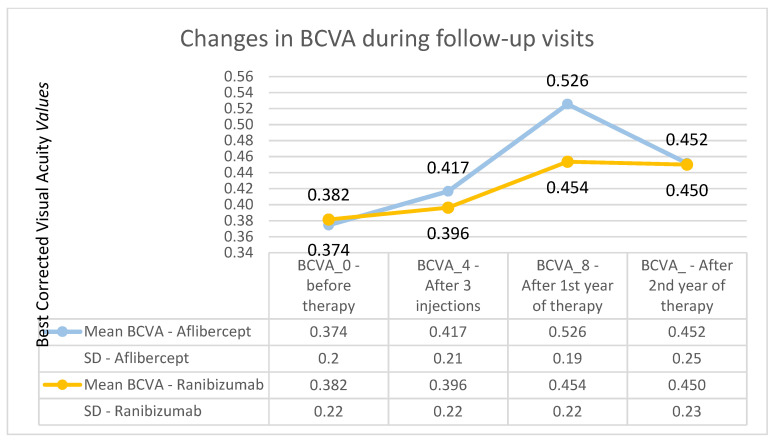
Changes in the best corrected visual acuity during individual follow-up visits—comparison of Aflibercept and Ranibizumab treatment results. The course of the best corrected visual acuity curve is much flatter for the group receiving Ranibizumab.

**Figure 4 medicina-57-01293-f004:**
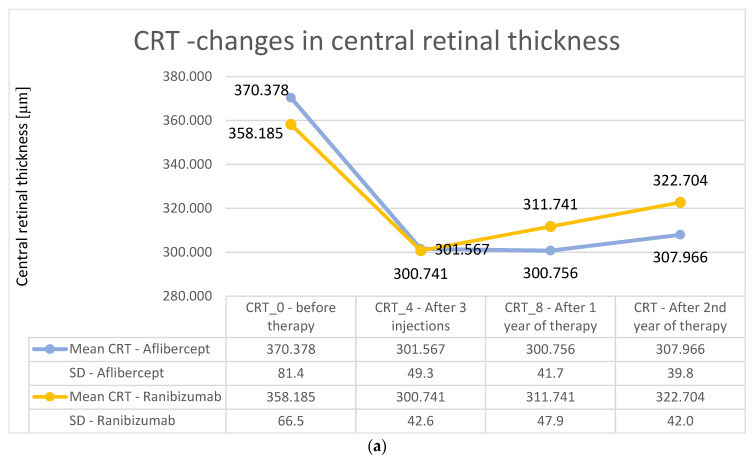
(**a**) Changes in the central retinal thickness during individual follow-up visits—comparison of treatment results for Aflibercept and Ranibizumab. The obtained decreases in retinal thickness did not differ significantly statistically between the two groups, (*p* > 0.05). (**b**) Average change in central retinal thickness after the first and second years of treatment—comparison of treatment results for Aflibercept and Ranibizumab. No significant statistical differences (*p* > 0.05) were observed.

**Figure 5 medicina-57-01293-f005:**
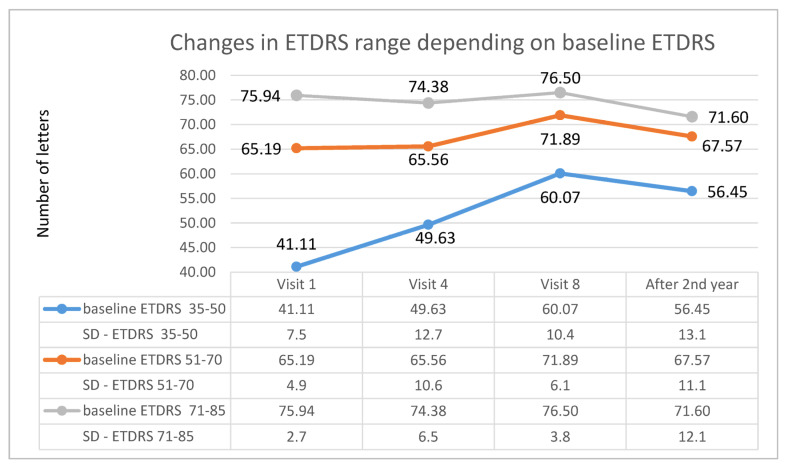
Changes in best corrected visual acuity in the ETDRS during individual follow-up visits in groups based on initial visual acuity—data for Aflibercept.

**Figure 6 medicina-57-01293-f006:**
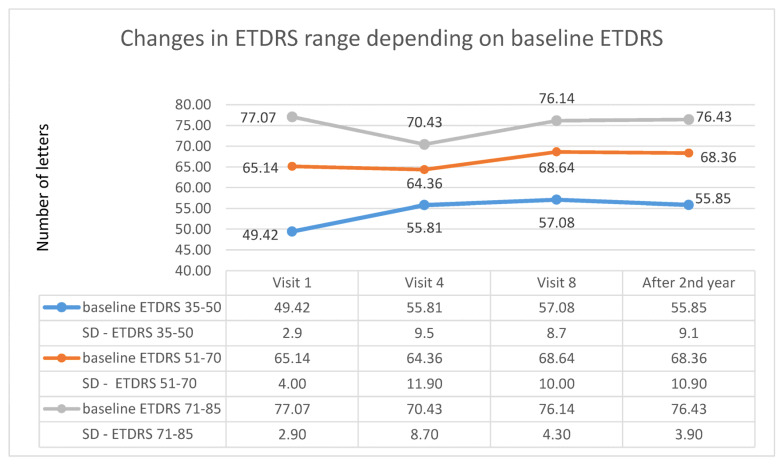
Changes in best corrected visual acuity in the ETDRS during individual follow-up visits in groups based on the initial visual acuity—data for Ranibizumab.

**Table 1 medicina-57-01293-t001:** Characteristics of the groups in terms of gender.

Anti-VEGF Patients		Frequency	Percent	Cumulative Percentage
Aflibercept	female	50	55.6	55.6
	male	40	44.4	100.0
	all	90	100.0	
Ranibizumab	female	26	48.1	48.1
	male	28	51.9	100.0
	all	54	100.0	

**Table 2 medicina-57-01293-t002:** Characteristics of the groups in terms of age.

Descriptive Statistics						
		*n*	Minimum	Maximum	Average	Standard Deviation
Aflibercept	Age	90	57	89	74.38	7.63
Ranibizumab	Age	54	52	96	75.72	8.85

**Table 3 medicina-57-01293-t003:** Characteristics of study groups in terms of baseline visual acuity (BCVA).

Descriptive Statistics						
		*n*	Minimum	Maximum	Average	Standard Deviation
Aflibercept	ETDRS_0—before therapy	90	35.0	80.0	59.9	14.08
Ranibizumab	ETDRS_0—before therapy	54	35.0	80.0	60.7	12.18

**Table 4 medicina-57-01293-t004:** Characteristics of study groups in terms of baseline central retina thickness (CRT).

Descriptive Statistics						
		*n*	Minimum	Maximum	Average	Standard Deviation
Aflibercept	CRT_0—before therapy	90	251.0	644.0	370.4	81.39
Ranibizumab	CRT_0—before therapy	54	236.0	569.0	358.2	66.51

## Data Availability

All the data are available from the corresponding author upon reasonable request.

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
