# Peer review of "Two Year Study of Aflibercept and Ranibizumab Intravitreal Therapy in Patients with Wet AMD"

_medicina, 2021, doi:10.3390/medicina57121293_

Round 1
Reviewer 1 Report
In this study by Luksa et al., the authors compare the outcomes of Aflibercept or Ranibizumab treatments in two separate groups of AMD patients in a two year long study. The patients were on a fixed regime during the first year- a total of seven injections; and a pro re nata regime in the second year with an average of above 3 injections in both groups. The authors measured visual acuity (using ETDRS) and central retina thickness (using OCT) 4 times- before starting the drug regime, at 3 months, 12 months and 24 months. This is the first report where a long term follow up, of 24 months, has been done to compare the effects of Aflibercept or Ranibizumab on visual acuity and central retina thickness in AMD patients. However, the main finding of the study that Aflibercept treatment and outcome is slightly better than Ranibizumab, over a long period of time, has been reported previously in several studies.
General comments-
- The title of the paper needs to be edited, it can be changed to “Two years study of Aflibercept and Ranibizumab intravitreal therapy in patients with wet AMD”
- The referencing style looks odd and should be checked with journal requirements.
- The results that Aflibercept is slightly better than Ranibizumab after one year therapy are consistent with some of the previously published results. Many of these references are missing and should be included and discussed in the Discussion- Kumar et al., 2013 (Retina), Fauser and Muether, 2015 (British journal of Ophthalmology), Heussen et al., 2014 (Graefe’s archive for clinical and experimental ophthalmology). Some groups have also reported no differences between Aflibercept and Ranibizumab after one year- Bohni et al., 2015 (BMC Ophthalmology). Please elaborate more in the discussion why there could be similarities or differences with these studies.
- In the figures, it is odd that the first figure is Figure 2. Figure 1 is missing. Please correct this. All the graphs in the figures are missing the standard deviation bars for all data points. Y-axis labels are also missing.
- This is optional, but it would be great if some OCT pictures, if available, can be included to show the retina thickness changes over the course of two years.
Specific comments-
- Lines 78: should include ‘..2011 and 2012 in USA and Europe respectively’
- Lines 141-143: please elaborate how this conversion from Snellen to ETDRS was done; if possible, average values for Snellen chart could be given too for the patients.
- Lines 166-169: ‘The analysis with Mann-Whitney….not statistically significant’ should go the Results section.
- Line 194: I believe this data is for group A, please mention.
- Lines 196 and 200: provide p-values for Wilcoxon test.
- Figures 2 and 3 can be combined into one figure, it will be easy to compare the two drugs.
- Lines 237-238: There is a sudden switch to the brand names of the drugs; the brand names should be mentioned somewhere towards the beginning, probably in the introduction.
- Lines 251-252: please show the correlation data for the results mentioned here.
- Figure 6.1 is essentially a repetition of figures 5.1, 5.2. Figures 5.1 and 5.2 should be removed, figure 6.1 is a better representation of the same data.
Author Response
Good morning,
Thank you very much for all your comments. We have tried to follow each of them. Among other things we have adopted the proposed new article title and improved the appearance of the bibliography. We have added information on what data are presented in the graphs and presented some figures that were originally omitted in the article.
Again, patients are being removed if vision drops below 0.2 for more than 2 months. So the study is being biased toward those who did well which can be alright if there are statistics on these patients as well in each of the groups.
This is because we only examined the outcomes of patients who were present in the drug program. Patients excluded from the drug program did not have the need for such frequent follow-up visits.
Reviewer 2 Report
I made many comments on the manuscript though most of them are for small stylistic errors. You include too many places in your statistics and I would alter your figure captions a bit. I do think you fail to address the real issue of the difference between the number of patients lost for VA decrease and other reasons in the Aflibercept group vs the Ranubizumab group and I think at least a sentence on this, qualifying your results ought to be done. I was also a bit unclear in a couple of places which generally later became clear but I think these issues could have been avoided and comments were made in those places in the paper.

Author Response

(The authors gave the same response as above.)

Round 2
Reviewer 1 Report
Most of the previous comments have been answered and appropriate corrections made to the manuscript.
The graphs in all the figures still lack the error bars (standard deviation or SEM), please include them in the graphs.
Author Response
Thank you for your messages. We have corrected the tables, please check that everything is now correct.